# Rendering Maxwell Equations into the Compressible Inviscid Fluid Dynamics Form

Peter Vadasz

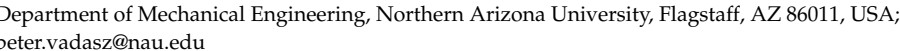

Department of Mechanical Engineering, Northern Arizona University, Flagstaff, AZ 86011, USA;
peter.vadasz@nau.edu

**Abstract:** Maxwell equations governing electromagnetic effects are being shown to be equivalent to the compressible inviscid Navier–Stokes equations applicable in fluid dynamics and representing conservation of mass and linear momentum. The latter applies subject to a generalized Beltrami condition to be satisfied by the magnetic field. This equivalence indicates that the compressible inviscid Navier–Stokes equations are Lorentz invariant as they derive directly from the Lorentz-invariant Maxwell equations subject to the same Beltrami condition, provided the pressure wave propagates at the speed of light, i.e., $v_o = c_o$. In addition, the derivation and results provide support for the claim that electromagnetic potentials have physical significance as demonstrated by Aharonov–Bohm effect, and are not only a convenient mathematical formulation.

**Keywords:** Maxwell equations; Navier–Stokes equations; inviscid flow; compressible flow; fluid dynamics; electromagnetism; Aharonov–Bohm effect





## 1. Introduction

Electromagnetic effects are related to fluid dynamics via applications of Magneto-Hydro-Dynamics (MHD) to a variety of fields, such as liquid metals [1–4], nanofluids [5,6], non-Newtonian fluids [7], and many others.

Electromagnetic phenomena, such as electromagnetic waves propagate in space following the solution of Maxwell equations. The latter is a set of four partial differential equations for the unknown variables of the electric field, magnetic field, charge density, and current density (charge flux). The solution to this set of equations is typically obtained via a gauge theory, i.e., introducing a scalar and a vector potential related to the electric and magnetic fields and solving the resulting wave equations for these potentials. Once the solution for the potentials is obtained a reverse transformation leads to the solution to the electric and magnetic fields. This procedure is similar to introducing scalar and vector potentials in attempting to solve fluid dynamics problems, although the latter are typically much more difficult. Also, the introduction of a stream function for solving two-dimensional fluid dynamics flows is also a similar procedure, although in this case much simpler than the electromagnetic one. In all these cases the introduced potentials are defined up to an integrating constant or for a vector potential, up to a gradient of an arbitrary function. The latter has no significance as long as the physically significant variables are the electromagnetic fields and not the potentials. Therefore, the introduction of the potentials was seen as a convenient mathematical solution method and the potentials themselves were not given any physical interpretation. Aharonov–Bohm effect (Aharonov and Bohm [8,9]) which was confirmed both theoretically as well as experimentally suggested the opposite, i.e., that the potentials do have physical significance, although no explanation for the latter nor the precise physical meaning of these potentials was provided.

Vadasz [10] showed that a continuous mass distribution for a general variable gravitational field $\mathbf{g}(t, \mathbf{x})$ is equivalent to a form identical to Maxwell equations from electromagnetism, subject to a modified Beltrami condition. Attempts at deriving equations that are

identical to Maxwell equations for continuous media have been presented particularly with application to fluid dynamics. For example, Marmanis [11] used an equation derived by Lamb [12] from the incompressible Navier–Stokes equations and uses it in deriving a new theory of turbulence. A similar approach was used by Sridhar [13] in order to "formulate the problem of advection and diffusion of a passive tracer by an arbitrary, incompressible velocity field", to find that the "problem is identical to the diffusive dynamics of a charged particle in electromagnetic fields constructed from the velocity field". Rousseaux et al. [14] tested experimentally and theoretically the concept of "hydrodynamic charge" in the case of a "coherent structure such as the Burgers vortex". These attempts apply to the incompressible fluid Navier–Stokes equations without the gravitational field and result in a form identical to Maxwell equations having the following correspondence: electromagnetic vector potential converts into velocity, magnetic field converts into vorticity, electric field converts into Lamb vector ($\mathbf{l} = -\boldsymbol{v} \times \nabla \times \boldsymbol{v}$), where $\boldsymbol{v}(t, \mathbf{x})$ is the velocity, and the electric charge converts into a "hydrodynamic charge" $q_H$ identical to the divergence of the Lamb vector, i.e., $q_H = \nabla \cdot \mathbf{l}$.

The present paper shows that Maxwell equations in free space governing electromagnetic phenomena are equivalent to the compressible inviscid Navier–Stokes equations subject to a generalized Beltrami condition. Consequently, a clear explanation of what physical meaning could be associated with these potentials is provided at the end of the paper.

## 2. Governing Equations

The following derivations use the definition of the mass-to-charge density ratio, assumed constant and assumed to take a linear form such as

$$\beta_q = \frac{\rho}{|\rho_q|} = \frac{m_q}{|q|} = \text{const.} [\text{kg/C}] \tag{1}$$

where $\rho$ is the mass density $[\text{kg/m}^3]$ related to the total mass $m_q$, and $\rho_q \ [\text{C/m}^3]$ is the electric charge density related to the electric charge $q$.

Then the definition of the electromagnetic momentum density (i.e., electromagnetic momentum per unit volume) is introduced in the form $\rho_q \boldsymbol{A}/\beta_q$ carrying units of $[\text{C/m}^2\text{s}]$, and where the vector $\boldsymbol{A} \ [N/A]$ is related to the magnetic field $\mathbf{B} \ [\text{T}]$ by the relationship

$$\mathbf{B} = -\nabla \times \mathbf{A} [\text{T}] \tag{2}$$

The electromagnetic momentum density will be shown to be identical to the current density (charge flux) $J_q = \rho_q \mathbf{A}/\beta_q \ [\text{C/m}^2\text{s}]$. Equation (2) produces the Gauss law for the magnetic field expressed in the form

$$\nabla \cdot \mathbf{B} = 0 \tag{3}$$

because the divergence of the curl of any vector is always zero.

Then, we use the Coulomb law in field form together with the Ampere law as follows:

$$\nabla \cdot \mathbf{E} = \frac{1}{\varepsilon_o} \rho_q \tag{4}$$

$$c_o^2 \nabla \times \mathbf{B} = \frac{1}{\varepsilon_o} \rho_q \frac{\mathbf{A}}{\beta_q} + \frac{\partial \mathbf{E}}{\partial t} \tag{5}$$

where $\mathbf{E}$ is the electric field in units of $[\text{N/C}]$, $\varepsilon_o$ is the permittivity of vacuum in units of $[\text{F/m}]$, $t$ is time in units of $[s]$, and $c_o$ is the speed of light in free space. The Faraday law of induction is presented in the form

$$\nabla \times \boldsymbol{E} = -\frac{\partial \boldsymbol{B}}{\partial t} \tag{6}$$

Equations (3)–(6) form the Maxwell equations governing electromagnetic phenomena in free space.

### 3. Converting the Governing Equations into a Fluid Dynamics Form

Applying the divergence operator on Equation (5) yields

$$0 = \frac{1}{\varepsilon_o} \nabla \cdot \left( \rho_q \frac{\mathbf{A}}{\beta_q} \right) + \frac{\partial}{\partial t} (\nabla \cdot \mathbf{E}) \tag{7}$$

Substituting (4) into (7) leads to

$$\frac{\partial \rho_q}{\partial t} + \nabla \cdot \left( \rho_q \frac{\mathbf{A}}{\beta_q} \right) = 0 \tag{8}$$

Equation (8) represents the conservation of electric charge, or the electric charge continuity equation while $\mathbf{A}/\beta_q$ represents the electric charge velocity.

Also, the following equation for the conservation of the electromagnetic momentum leads directly to Faraday law of induction (6), as follows

$$\rho_q \left[ \frac{\partial \mathbf{A}}{\partial t} + (\mathbf{A} \cdot \nabla) \mathbf{A} \right] = -c_o^2 \beta_q \nabla \rho_q + \rho_q \mathbf{E} \tag{9}$$

Dividing Equation (9) by $\rho_q$ produces

$$\frac{\partial \mathbf{A}}{\partial t} + (\mathbf{A} \cdot \nabla) \mathbf{A} = -\nabla \left[ s_q c_o^2 \beta_q \ln |\rho_q| \right] + \mathbf{E} \tag{10}$$

where $s_q = q/|q| = +1$ if $q > 0$ & $-1$ if $q < 0$. By using the following identity

$$(\mathbf{A} \cdot \nabla) \mathbf{A} = \frac{1}{2} \nabla (\mathbf{A} \cdot \mathbf{A}) - \mathbf{A} \times (\nabla \times \mathbf{A}) \tag{11}$$

into Equation (10) yields

$$\frac{\partial \mathbf{A}}{\partial t} = -\nabla \left[ s_q c_o^2 \beta_q \ln |\rho_q| + \frac{1}{2} (\mathbf{A} \cdot \mathbf{A}) \right] + \mathbf{A} \times (\nabla \times \mathbf{A}) + \mathbf{E} \tag{12}$$

The term $\Phi = s_q c_o^2 \beta_q \ln |\rho_q| + \frac{1}{2} (\mathbf{A} \cdot \mathbf{A})$ can be seen as a reduced pressure which is equivalent to electromagnetic potentials.

Taking the curl ($\nabla \times$) of Equation (12) and using Equation (2) leads to

$$-\frac{\partial \mathbf{B}}{\partial t} = \nabla \times \mathbf{E} - \nabla \times [\mathbf{A} \times \mathbf{B}] \tag{13}$$

Subject to satisfying the following generalized Beltrami condition (Rousseaux et al. [14], Yoshida et al. [15], Mahajan and Yoshida [16], Gerner [17], Amari et al. [18], Bhattacharjee [19], Lakhatakia [20])

$$\nabla \times [\mathbf{A} \times \mathbf{B}] = 0 \tag{14}$$

Equation (13) converts into the Faraday law of induction in the form

$$-\frac{\partial \mathbf{B}}{\partial t} = \nabla \times \mathbf{E} \tag{15}$$

The generalized Beltrami condition (in fluid dynamics) is satisfied when one of the following occurs:

$$\text{(a) } \mathbf{B} = -\nabla \times \mathbf{A} = 0 \tag{16}$$

Then the flow is irrotational, and in electromagnetism, it implies no magnetic field.

$$(b) \ \mathbf{A} \times \mathbf{B} = 0 \tag{17}$$

i.e., the electromagnetic vector potential **A** and the magnetic field **B** are parallel (Beltrami condition). In this case **A** cannot be two-dimensional.

$$(c) \ \nabla \times [\mathbf{A} \times \mathbf{B}] = 0 \tag{14}$$

which is the generalized Beltrami condition implying the existence of a scalar potential $\psi$ such that

$$\mathbf{A} \times \mathbf{B} = \nabla \psi \tag{18}$$

Satisfying identically the generalized Beltrami condition (14). This scalar potential can be in particular (not necessarily)

$$\psi = -\frac{1}{2}\mathbf{A} \cdot \mathbf{A} \tag{19}$$

in which case (18) converts by using (2) and (19) into

$$\mathbf{A} \times (\nabla \times \mathbf{A}) = \nabla\left(\frac{1}{2}\mathbf{A} \cdot \mathbf{A}\right) \tag{20}$$

In all these cases the term $(\mathbf{A} \cdot \nabla)\mathbf{A}$ in Equations (9) and (10) vanishes, except for its effect on the reduced pressure term in Equation (12).

Equations (8) and (9) that emerged directly from the Maxwell equations have a form identical to the compressible inviscid Navier–Stokes equations from fluid dynamics with the following equivalence:

$\rho \to \rho_q$, $v \to \mathbf{A}/\beta_q$, $v_o^2 \to c_o^2$, $\mathbf{g} \to \mathbf{E}$, $\boldsymbol{\xi} = -\nabla \times v \to \mathbf{B} = -\nabla \times \mathbf{A}$, where $v$ is the velocity vector, $\boldsymbol{\xi}$ is the vorticity vector, $g$ is the variable gravitational field, and $v_o$ is the speed of propagation of the pressure wave $p$, i.e., by using a linear constitutive relationship between pressure and mass density:

$$p = p_o + v_o^2(\rho - \rho_o) \tag{21}$$

Therefore if the vector potential **A** is linearly related to the electric charge velocity in the form

$$v = \frac{\mathbf{A}}{\beta_q} \tag{22}$$

identifying the electromagnetic momentum $\rho_q \mathbf{A}/\beta_q$ to the electric current density (charge flux) $\mathbf{J_q} \ [\mathrm{A/m^2}]$, i.e.,

$$\mathbf{J_q} = \rho_q v = \rho_q \frac{\mathbf{A}}{\beta_q} \tag{23}$$

and assuming a linear relationship between the mass density and the charge density, such as the one presented in Equation (3), and if the generalized Beltrami condition (14) is satisfied, then by using (21), Equations (8) and (9) become

$$\frac{\partial \rho}{\partial t} + \nabla \cdot (\rho v) = 0 \tag{24}$$

$$\rho\left[\frac{\partial v}{\partial t} + (v \cdot \nabla)v\right] = -\nabla p + \rho_q \mathbf{E} \tag{25}$$

Equations (24) and (25) are the compressible inviscid Navier–Stokes equations for a charged continuum identical to a fluid. They were derived directly from the Maxwell equations subject to the generalized Beltrami condition and assuming linear relationships

between the electromagnetic vector potential **A** and charge velocity $v$, and between the mass density $\rho$ and charge density $\rho_q$.

This result may be linked to the Aharonov–Bohm effect (Aharonov and Bohm [8,9]), which conceptually challenges the view that expressing the Maxwell equations in terms of potentials and hence converting them into a gauge theory is only a mathematical reformulation with no physical consequences because the scalar and vector potentials have no apparent physical significance. The result presented in the present paper supports Aharonov and Bohm's [8,9] claim of the physicality of electromagnetic potentials $\Phi = s_q c_o^2 \beta_q \ln|\rho_q| + \frac{1}{2}(\mathbf{A} \cdot \mathbf{A})$ and **A** by illustrating that the former is related to a reduced pressure, while the latter is related to the electric charge velocity, as shown above.

## 4. Conclusions

The Maxwell equations were shown to convert into the compressible inviscid Navier–Stokes equations subject to the magnetic field satisfying a generalized Beltrami condition. Since Maxwell equations are Lorentz invariant, the latter suggests that subject to the same condition the compressible inviscid Navier–Stokes equations are Lorentz invariant too provided the pressure wave propagates at the speed of light, i.e., $v_o = c_o$. Finally, the results also support the claim that electromagnetic potentials have physical significance as demonstrated by Aharonov–Bohm effect, and are not only a convenient mathematical formulation.

**Funding:** This research received no external funding.

**Data Availability Statement:** There is no data created.

**Conflicts of Interest:** The author declares no conflict of interest.

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
