# Peer review of "Rendering Maxwell Equations into the Compressible Inviscid Fluid Dynamics Form"

_fluids, doi:10.3390/fluids8110284_

Round 1

Reviewer 1 Report

Analogy is interesting, but there exists a description in the section of Conclusion which cannot be acceptable from physical point of view. The following sentence in the section "4. Conclusions" must be deleted:

"the compressible inviscid Navier-Stokes equations are Lorentz invariant too"

because the speed v_o is not the light speed, but the sound speed. speed. Therefore, it is not appropriate to mention the Lorentz invariance in the conclusion.

Minor comments: There are two parts in the manuscript that the equation (2) is mis-cited as equation (1): one in the section 2 and the other in 3.

Author Response

SEE ATTACHED PDF FILE.

Reviewer 2 Report

The article explores a connection between Maxwell's equations, which govern electromagnetic phenomena, and the compressible inviscid Navier-Stokes equations, which are used in fluid dynamics to describe mass conservation and linear momentum. This connection reveals that under a generalized Beltrami condition for the magnetic field, these two sets of equations are equivalent. Importantly, this equivalence highlights that the compressible inviscid Navier-Stokes equations are Lorentz invariant, as they directly stem from the Lorentz invariant Maxwell equations, both subject to the same Beltrami condition. The finding lends support to the idea that electromagnetic potentials have real physical significance, as evidenced by phenomena like the Aharonov-Bohm effect, rather than being purely mathematical constructs.

1.       However, there are lack of testing the derived equation in order to prove that it is accurate for an analytical investigation. Hence, it is not a research based paper, just a derivation for some methodologies.

2.       No results presentation including graph, data and figure to show the relationship works well and in line with other approximation.

3.       The equations are not well written, low quality of presentation.

4.       There is no indication that the novelty is achieved.

5.       Hence, this article is rejected for a publication. 

·         However, there are lack of testing the derived equation in order to prove that it is accurate for an analytical investigation. Hence, it is not a research based paper, just a derivation for some methodologies.

·         No results presentation including graph, data and figure to show the relationship works well and in line with other approximation.

·         There is no indication that the novelty is achieved.

Moderate. 

Author Response

SEE ATTACHED PDF FILE

Reviewer 3 Report

See the attachment.

Author Response

SEE ATTACHED PDF FILE.

Reviewer 4 Report

This is a very useful paper

Author Response

SEE ATTACHED PDF FILE.

Reviewer 5 Report

The author have presented that " Maxwell equations governing electromagnetic effects are being shown to be equivalent to the compressible inviscid Navier-Stokes equations applicable in fluid dynamics and representing conservation of mass and linear momentum.' It is really a very nice work performed by the author. 

Author Response

SEE ATTACHED PDF FILE.

Round 2

Reviewer 2 Report

All considerations are been noted

Reviewer 3 Report

The authors have addressed my questions and I would like to recommend it  for publication.